**Data Availability Statement:** Data are owned by the University of Nigeria and ethical restrictions have been imposed by the Health Research Ethics

# Occupational biopsychosocial factors associated with neck pain intensity, neck-disability, and sick leave: A cross-sectional study of construction labourers in an African population

**Chinonso N. Igwesi-Chidobe** [1,2]*, **Excellence Effiong**[2], **Joseph O. Umunnah**[3], **Benjamin C. Ozumba**[2,4]

**1** School of Allied Health Professions and Midwifery, Faculty of Health Studies, University of Bradford, Bradford, United Kingdom, **2** Global Population Health (GPH) Research Group, University of Nigeria, Nsukka, Nigeria, **3** Department of Medical Rehabilitation, Nnamdi Azikiwe University, Nnewi Campus, Nnewi, Anambra State, Nigeria, **4** Faculty of Medicine, College of Medicine, University of Nigeria, Enugu Campus, Enugu, Nigeria

* c.igwesi-chidobe@bradford.ac.uk, chinonso.chidobe@unn.edu.ng

## Abstract

### Introduction

The burden and impact of neck pain is high in African countries including Nigeria. This study investigated the occupational biomechanical and occupational psychosocial factors associated with neck pain intensity, neck disability and sick leave amongst construction labourers in an urban Nigerian population.

### Methods

This cross-sectional study measured clinical neck pain outcomes, occupational biomechanical factors, and occupational psychosocial factors. Descriptive, and univariate/multivariate inferential statistical analyses were conducted.

### Results

Significant independent factors associated with neck pain intensity were order and pace of tasks being dependent on others (β = 0.35; p<0.0001); inability to take breaks in addition to scheduled breaks (β = 0.25; p<0.0001); inability to work because of unexpected events (β = 0.21; p<0.0001); inability to control the order and pace of tasks (β = 0.20; p<0.0001); and weight of load (β = 0.17; p<0.0001); accounting for 53% of the variance in neck pain intensity. Significant independent factors associated with neck disability were weight of load (β = 0.30; p<0.0001); duration of load carriage (β = 0.16; p = 0.01); working under time pressure/ deadlines (β = 0.16; p = 0.02); and accounting for 20% of the variance in neck disability. Significant independent factor associated with sick leave was duration of load carriage (β = 0.15; p = 0.04), in a non-significant regression model explaining -4% of the variance in sick leave. Addition of pain intensity significantly explained more variance in neck disability

Committee of the University of Nigeria Teaching Hospital. In line with the recommendations of the Health Research Ethics Committee of the University of Nigeria Teaching Hospital, data supporting this study can be obtained via reasonable request to the Head of the Health Research Ethics Committee of the University of Nigeria Teaching Hospital through email (gilbert.adimora@unn.edu.ng) or to the corresponding author of this paper.

**Funding:** The lead author was partly supported with personal grant from the 2020 Nature Research Award for Driving Global Impact. The funders had no role in study design, data collection and analysis, decision to publish, or preparation of the manuscript.

**Competing interests:** The authors have declared that no competing interests exist.

(31.0%) but less variance in sick leave (-5%), which was not statistically significant (F (10, 190) = 0.902, p = 0.533).

## Conclusions

Occupational biomechanical factors may be more important than occupational psychosocial factors in explaining neck disability and sick leave. In contrast, occupational psychosocial factors may be more important than occupational biomechanical factors in explaining neck pain intensity in this population in Nigeria.

## Introduction

Globally, neck pain is the second most common musculoskeletal cause of disability-adjusted life-years (DALYs) in the working population, behind low back pain (LBP) [1]. Neck pain (defined as 'activity-limiting neck pain with or without radiation to the upper limbs lasting for at least 24 hours) is a major public health concern globally due to significant direct and indirect individual and societal costs, and the negative impact on overall health, function, well-being and quality of life [2, 3]. There is a wide variation in the overall prevalence of neck pain in the general population ranging from 0.4% to 86.8% (mean: 23.1%) [4]. Point prevalence rates range between 0.4% and 41.5% (mean = 14.4%); with 1 year prevalence rates ranging between 4.8% and 79.5% (mean = 25.8%) [3]. As of 2019, neck pain had an age-standardised prevalence rate of 27.0 per 1000 population [4].

The prevalence of spinal disorders, particularly back and neck pain is an increasing concern in low- and middle-income countries (LMICs) [5]. Reasons for the concern in these countries include the ageing population, increasing prevalence of spinal pain, high poverty levels combined with a high burden of infectious diseases in environments with limited health resources and health information [6–11]. The consequences of spinal pain are potentially even more devastating in sub-Saharan African countries due to maladaptive pain beliefs, dangerous living and working conditions, plus high poverty levels and limited health services [8, 12, 13]. In Nigeria, studies have investigated the prevalence of neck pain amongst specific population groups including university students, dentists, and rural farmers with prevalence rates ranging from 10% to 82% [14–16]. The construction industry is associated with a high burden of neck pain-related injury in high-income countries [17]. The burden of neck pain in low and middle-income countries including Nigeria [18–20] may be associated with nearly non-existent occupational health policies and dangerous working conditions. However, studies have not investigated the biomechanical factors alongside psychosocial factors associated with multiple neck pain outcomes in any population group including construction workers in Nigeria.

A previous study investigated the biomechanical and psychosocial factors associated with disability amongst people with chronic LBP in rural Nigeria [8]. The results showed that psychosocial factors were the most important factors associated with both self-reported and performance-based disability, explaining 62.5% of self-reported disability and 49.1% of performance-based disability [8]. Unexpectedly, occupational biomechanical factors were not associated with self-reported or performance-based disability [8] which contradicted the findings from the qualitative studies [12, 13] in this population. However, the rural Nigerian population studied were all peasant farmers either on a full-time or part-time basis, which might have obfuscated associations with occupational biomechanical factors [8, 12]. The involvement of mostly farmers might have implied that a higher value of occupational biomechanical

factors might have been reflecting people still in active farming. Conversely, lower biomechanical factors might have been reflecting those who were no longer farming in that population-based cross-sectional study that did not measure sick leave [8]; which was also suggested in qualitative studies conducted in this population [12, 13]. This has been termed 'healthy worker' effect [21]. Furthermore, the use of total scoring (rather than individual items) of the occupational risk factor questionnaire (ORFQ) may have obscured the relevance of occupational biomechanical factors in explaining chronic LBP disability [8]. This could be due to different individuals having different aggravating biomechanical factors. For instance, the aggravating biomechanical factors in some individuals may be protective in some other individuals. A total score of biomechanical exposure could therefore cancel out the impact of individual biomechanical factors which might explain the limited relevance of biomechanical factors in explaining chronic LBP disability in that study [8]. This underscores the importance of biomechanical exposure outcome measures that can capture this level of nuanced measurement. Moreover, the fact that participants with limited literacy found the questionnaire items difficult to understand [22] may have contributed to the lack of relevance of biomechanical factors in predicting chronic LBP in that study [8]. These previous results could also simply imply that occupational biomechanical factors are not important in explaining chronic LBP disability as suggested in that study [8] but may be important in other spinal pain outcomes. The latter is supported by a more recent study in Nigeria suggesting that occupational biomechanical factors may be the strongest independent factors associated with a different spinal pain outcome–a current episode of LBP [23]. The utilization of a total scoring of biomechanical exposure (ORFQ) in that study prohibited an understanding of the relative importance of individual biomechanical factors. Furthermore, the lack of any objective measurement of biomechanical exposure in that study increased the risk of recall bias [23].

Evidence suggests that biomechanical factors including sustained flexion and rotation, and spinal loading [24–30], and job-related psychosocial factors including work pressure and stress [24, 25, 31, 32] may be associated with spinal pain outcomes including pain intensity, disability and sick leave. Previous evidence from a rural Nigerian population suggests that spinal pain intensity may be an independent predictor of disability [8]. These factors informed the proposed theoretical model depicted in Fig 1 below which underpins this study.

This model proposes that biomechanical and psychosocial factors would each be independently associated with neck pain intensity, neck disability and sick leave. Furthermore, the model proposes that neck pain intensity would be independently associated with each of the other dependent neck pain outcomes–neck disability and sick leave. This study will therefore clarify the findings from the previous studies and further improve an understanding of the factors driving different spinal pain outcomes in Nigeria. It was important to investigate other populations such as urban Nigerian population groups with other common spinal pain conditions apart from back pain such as neck pain. Utilization of other measures of exposure to biomechanical factors that include an objective component–the actual weight of the load, rather than a complete dependence on the perception of weight, and separating the different categories of biomechanical exposure rather than using a total score, might clarify the importance of specific biomechanical factors for specific spinal pain outcomes in this population. Including a broader range of outcomes apart from disability such as sick leave and pain intensity would help to clarify the importance of biomechanical factors relative to psychosocial factors for different spinal pain outcomes in Nigeria. In view of these, this study aimed to determine the occupational biomechanical and occupational psychosocial factors associated with neck pain intensity, neck-disability, and sick leave amongst construction labourers in an urban Nigerian population. This study is reported according to the guidelines in Strengthening the Reporting of Observational Studies in Epidemiology (STROBE) statement [33].

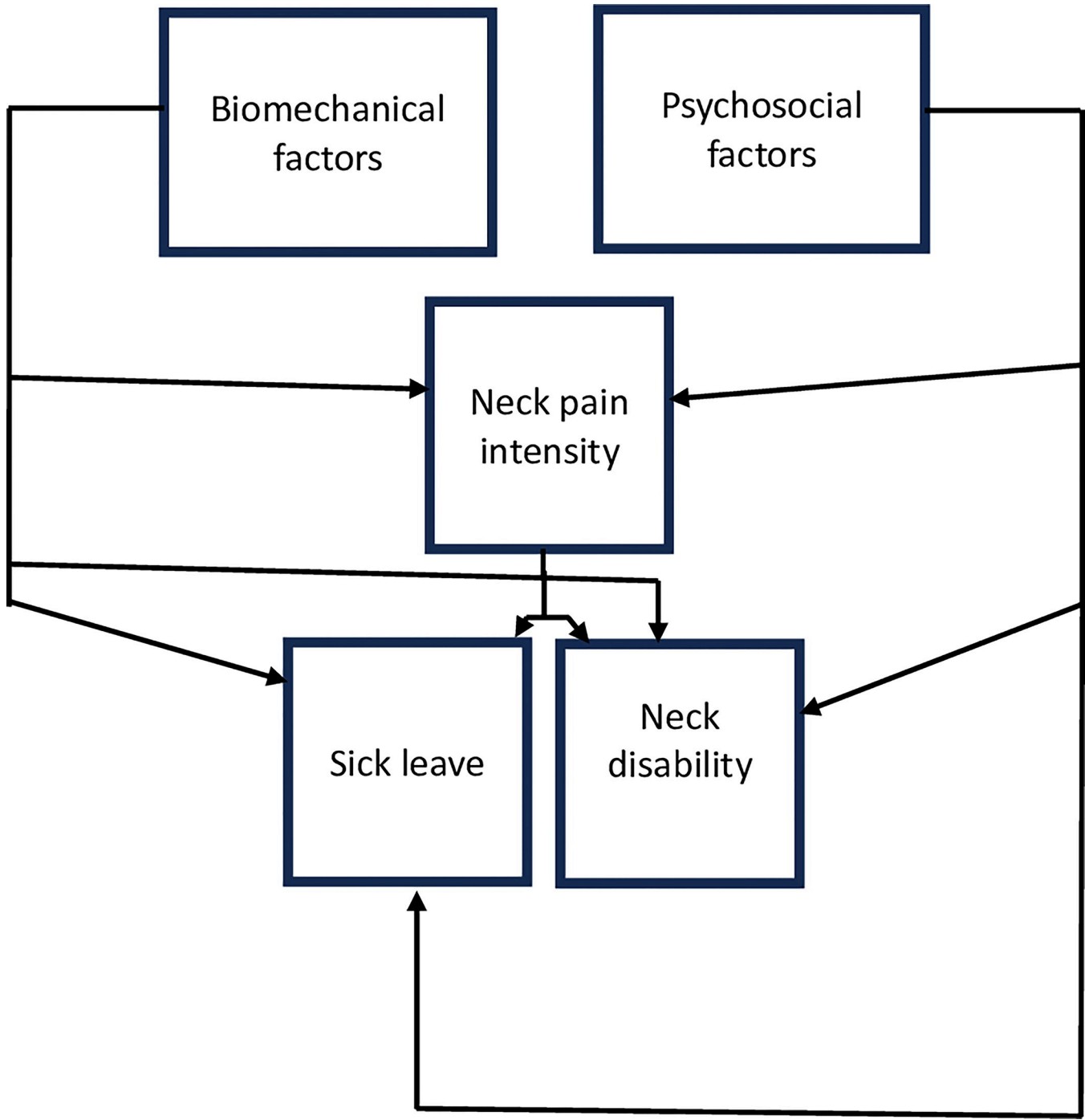

**Fig 1. The biopsychosocial theoretical model informing the study.**

## Materials and methods

### Ethical considerations

Ethical approval was obtained from the Research Ethics Committee of the University of Nigeria Teaching Hospital (Ref: UNTH/HREC/2021/01/13). Written informed consent was obtained from all participants.

## Study design

Cross-sectional study amongst construction labourers in an urban Nigerian setting.

## Study setting

This study took place in the three largest estates being constructed in Enugu metropolis between August 2021 and January 2022 with each estate having about 30 constructions sites. Attempts were made to recruit people from all 90 construction sites in the three estates. Enugu is the capital city of Enugu state, southeastern Nigeria. It has seventeen Local Government Areas (LGAs) with about 202 communities. Three of these LGAs–Enugu east, Enugu north, Enugu south are located in the metropolis [34]. The state is generally residential with few industries but having several estates and construction sites, making it a reliable location for this study.

## Sample size

A priori sample size calculation was performed with G power version 3.1 software [35]. A sample size of 189 would give a 95% power to detect a medium regression effect size ($f^2$) of 0.15 at alpha ($\alpha$) of 0.05 with tested predictors of 13 based on the biopsychosocial theoretical model of pain and research evidence that were previously explained. Thirteen independent variables were chosen to account for the 6 occupational biomechanical factors (for each participant), 5 occupational psychosocial factors overall, and a couple of socio-demographic factors that might have significant associations with each dependent outcome (neck pain intensity, neck disability or sick leave). Arrangement was made to recruit up to 201 participants to account for potential data loss due to incomplete or incorrect data collection.

## Participants and recruitment

The target population was construction labourers working in construction sites in Enugu metropolis, where construction labourers usually carry load such as concrete, sand, blocks on their head to assist building engineers in construction. Verbal and written information about the study were presented to eligible participants in all 90 construction sites in the three selected building estates. They were given 3 days to decide participation or not. Written informed consent was obtained. Eligible participants that decided to participate signed the consent forms or thumb printed on the consent forms and were then recruited using simple random sampling (the lottery method). This involved counting the number of interested eligible participants in all 90 construction sites. Numbers 1–486 (number that was interested and eligible) were written in separate pieces of paper which were folded, put into a box, and mixed. Lastly, each participant randomly selected a folded paper from the box. Those who chose the range of numbers within the sample size (1–201) were recruited into the study.

Participants were eligible if they were ≥18 years of age, and their construction labourer job typically involved carrying load on the head daily which is the norm in this population. Participants were excluded if they had congenital or acquired musculoskeletal deformity such as kyphosis, lordosis, scoliosis, or kyphoscoliosis; previous neck trauma, recent neck trauma including whiplash injury; a previous surgery around the neck; or they had other serious underlying pathologies such as malignancy, infection, fracture, spinal stenosis, or metabolic disorders.

## Variables and outcome tools

The socio-demographic variables for which data were collected included age, sex, and education. Patient-reported outcome measures were cross-culturally adapted and validated prior to use.

Pain intensity was measured with the Numerical Rating Scale-11 (NRS-11) which is a one-item, one-dimensional patient-reported scale used for assessing self-reported pain intensity [36]. The NRS consists of eleven numbers (0–10) with verbal descriptors representing the entire possible range of pain intensity with 0 indicating "no pain" and 10 indicating "maximum pain" [37]. NRS can be administered verbally or in a written format, is simple and easily understood, and is easy to administer and score [38]. The measure has excellent reliability and validity in this population [22, 39, 40].

Neck disability was assessed with the Neck Disability Index (NDI) which is a patient-reported questionnaire consisting of 10 items that measures neck pain related disability with items including personal care, lifting, reading, work, driving, sleeping, recreational activities, neck pain intensity, concentration and headache [41, 42]. Each item is scored on an incremental scale from 0 (no disability for the item) to 5 (maximum disability for the item) for a maximum total score of 50. The total score is reported as either out of 50 or as a percentage out of 100 with higher scores indicating greater self-reported disability due to neck pain. The severity of neck disability include 0–4 points (0–8%)–no disability, 5–14 points (10–28%)–mild disability, 15–24 points (30–48%)–moderate disability, 25–34 points (50–64%)–severe disability, 35–50 points (70–100%)–complete disability [42, 43]. NDI has excellent validity, reliability and is clinically responsive [44–46].

Occupational psychosocial factors were assessed with the first 5 items of the ORFQ [47]. The Igbo-version of items 1–5 of the ORFQ which measure work organizational factors such as work pressure and stress was used by scoring individual items, yes or no [8, 22]. These items were: (1) Can you usually take breaks in your job in addition to the scheduled breaks? (2) Do you often find that you cannot work because of unexpected events, such as machine break down or material not delivered? (3) Can you usually control the order and pace of your tasks? (4) Is the order and pace of your tasks usually dependent on others (machines, computers, customers)? (5) Do you usually work under time pressure and deadlines? Although appearing similar, item 3 'Can you usually control the order and pace of your tasks?' is different from item 4 'Is the order and pace of your tasks usually dependent on others (machines, computers, customers)?' Whilst item 3 focuses on an internal locus of control such as the ability of participants to decide which tasks to do first and which ones to do later at different work phases, item 4 focuses on an external locus of control such as the order and pace of participants' tasks being dependent on external factors such as machines, computers, customers etc [48]. During the field work, it was found that the labourers' order and pace of tasks was mainly dependent on the needs of the builder (construction Engineer). For instance, if the builder needed a heavy object, the labourers would provide this. The builders determined how many blocks the labourers would carry and the number of the bags of cement they would mix per day. Therefore, the two questions were aimed at distinguishing participants' perception of personal control in the order and pace of tasks e.g., the number of blocks or bags of cement each labourer mixed per hour or how long they worked to meet the builder's daily expectations or any other personal adaptations they could make to meet the builder's daily demands. Research suggests good validity and reliability of the psychosocial content of the tool [22, 47, 49, 50].

Exposure to occupational biomechanical factors was measured by combining objective and subjective procedures. Head load carriage history was assessed with a self-developed questionnaire with 3 items assessing the intensity (weight in kilograms), frequency (number of days per week), and duration (number of hours per day) of carrying a weight subjectively described as heavy by each participant which was then objectively measured with the weighing scale. The three questionnaire items were: a) Do you usually carry any load that you feel is heavy for you (described object is then weighed with a weighing scale)? b) How many days do you carry this load in a week? c) How many hours do you carry this load in a day? Weight of head load was

measured with a USB digital body weight scale of weighing range 0.1-180KG (Mitronicas Global). Neck posture history was assessed with the self-developed questionnaire with 3 items assessing predominant neck posture, frequency (number of days per week) of the neck in that posture, and duration (number of hours per day) of working with the neck in that position. The three items were: a) Do you often have to work with your neck bent forward (flexion), bent backwards (extension) or twisted (trunk rotation–rotating the trunk to either side) [select one predominant position]? b) How many days in a week do you keep your head in this predominant position? c) How many hours in a day do you assume this predominant position? The construct of biomechanical exposure measured by this questionnaire is evidence-based [47, 51–57] and the test-retest reliability was confirmed by a sub-sample of 50 participants (Intraclass Correlation Coefficient = 0.536; internal consistency [Cronbach's alpha] = 0.632).

Sick leave was measured with a single item in the self-developed questionnaire which asked how many days in the past four weeks participants have stayed off from work due to neck pain and disability, which aligns with the World Health Organisation Disability Assessment Schedule and is found to be valid and reliable in this population [40].

## Data collection procedures

The leader of each construction site was visited, and the study procedures and implications were explained to them. On getting their approval for the study, announcements were then made to labourers explaining the aims of the study and the procedures involved. The announcements also emphasized the voluntary nature of the study and the eligibility criteria. Informed consent was subsequently obtained from the construction labourers who indicated interest in participating in the study. They were then screened against the eligibility criteria. Eligible participants who were literate self-completed the patient-reported questionnaires whilst trained research assistants interviewer-administered the questionnaires for the participants with limited literacy. All questionnaires were collected on the same day they were given or administered to participants.

## Statistical analyses

Data were analysed with the Statistical Package for Social Sciences (IBM SPSS), version 23.0 using two-tailed analyses. Normality of data was investigated using the normal distribution curve to inform the descriptive summary statistics and the inferential statistics conducted. Frequencies and percentages, and median and interquartile ranges were used to summarise socio-demographic characteristics. Exposure to occupational biopsychosocial factors, and neck pain outcomes of neck pain intensity, neck disability and sick leave were summarized with means and standard deviations, median and interquartile ranges, and/or frequencies and percentages.

Bivariate associations between each of the dependent variables (neck pain intensity, neck disability or sick leave) and each of the socio-demographic variables were investigated. Spearman's correlation was used to find bivariate relationships between the dependent variables and the continuous socio-demographic variables–age. Mann-Whitney U was used to determine if the dependent variables varied according to the dichotomous categorical sociodemographic variables–sex. Kruskal-Wallis test was used to determine if the dependent variables differed according to the polychotomous categorical sociodemographic variables–educational status. The socio-demographic variables with associations at $p \leq 0.25$ with the neck pain outcomes (dependent variables) were entered into sequential multiple regression analyses to account for their effects. We used a p-value cut-off point of 0.25 for inclusion into the regression models, as the traditional cut-off level of 0.05 can fail in identifying variables known to be important [58–60].

Bivariate associations between each of the dependent variables (neck pain intensity, neck disability or sick leave) and each of the independent variables were investigated. Spearman correlation analysis was used to investigate the associations between the dependent neck pain outcomes (neck pain intensity, neck disability and sick leave) and the continuous independent variables–all occupational biomechanical factors except predominant neck posture. Mann-Whitney U was used to investigate if the dependent neck pain outcome variables differed according to the dichotomous categorical independent variables–all the occupational psychosocial factors. Kruskall Wallist test was used to determine whether the dependent neck pain outcome variables differed according to the polychotomous categorical independent variables–predominant neck posture which is one of the occupational biomechanical factors.

A bivariate correlation matrix was used for the initial investigation of multicollinearity between the independent biomechanical and psychosocial variables. Spearman correlation analysis was used to determine the relationship between all the independent biomechanical and psychosocial variables (categorical variables were dummy coded). Multicollinearity was said to be present when any resulting correlation coefficient was greater than 0.8 [61]. Multicollinearity was further investigated with tolerance and variance inflation factor (VIF) in three standard regression analyses inputting all the independent biomechanical and psychosocial variables for the three dependent variables–neck pain intensity, neck disability and sick leave. A tolerance close to 1 indicates very little multicollinearity, whereas a value close to zero suggests that multicollinearity may be a threat. A value of VIF exceeding 10 was interpreted as the presence of multicollinearity, with values above 2.5 regarded as a cause for concern [61, 62]. There are no formal cutoff values of tolerance and VIF for determining the presence of multicollinearity [61].

In line with the biopsychosocial theoretical model underpinning this study (which proposes that biomechanical and psychosocial factors would each be independently associated with neck pain intensity, neck disability and sick leave), all the occupational biomechanical and psychosocial variables were entered into three sequential multiple regression analyses with neck pain intensity, neck disability and sick leave as the criterion variables after controlling for any significant (p≤0.25) socio-demographic variables. Two additional sequential multiple regression analyses were conducted for neck disability and sick leave with pain intensity included as an independent variable. This aligns with the biopsychosocial theoretical model underpinning this study (which proposes that neck pain intensity would be independently associated with each of the other dependent neck pain outcomes–neck disability and sick leave). Alpha levels were set at 0.05.

## Results

Fig 2 below illustrates how participants were recruited at the different stages and the response rates.

Tables 1–3 summarize the socio-demographic characteristics of the participants, exposure to occupational biomechanical and psychosocial factors, and the dependent variables of pain intensity, neck disability and sick leave, respectively. Majority of the participants were male and had attended secondary school. The mean weight carried at once was 39kg for 9 hours per day and 6 days per week, with the neck predominantly in a twisted (rotated) position. Majority of the participants were exposed to all occupational psychosocial factors. The median pain intensity was 6 with majority having moderate neck disability but were only 1 day off work in the past 4 weeks.

Table 4 shows the bivariate associations between each of the socio-demographic characteristics, occupational biomechanical factors, occupational psychosocial factors, and each of the

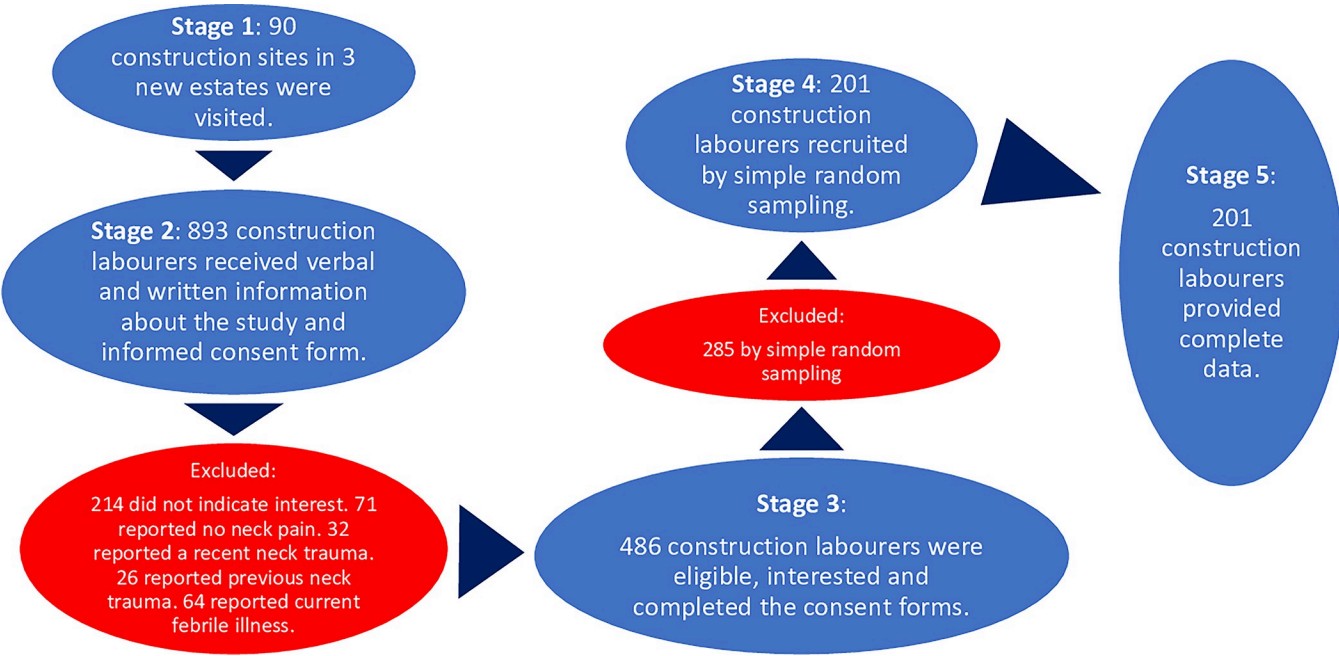

**Fig 2. Summary of the sampling and response rates.**

dependent variables (neck pain intensity, neck disability and sick leave), respectively. The significant (p≤0.25) sociodemographic factors associated with neck pain intensity were age and educational status. There was no significant (p≤0.25) sociodemographic factor associated with neck disability and sick leave.

Table 5 and S1 Table present the bivariate associations between each of the independent biomechanical and psychosocial variables, and the collinearity diagnostics for the three regression models, respectively. In Table 5, association between neck posture frequency and frequency of load carriage had a positive correlation coefficient ≥ 0.8; and the association between neck twisted (rotated) posture and duration of load carriage had a negative correlation coefficient ≥ 0.8. Furthermore, in S1 Table, VIF values of approximately 7 for frequency of load carriage and neck posture frequency in the three regression models indicate multicollinearity with the presence of frequency of load carriage and neck posture frequency. Therefore,

**Table 1. Sociodemographic characteristics of participants.**

| Socio-demographic variables n = 201 | Median (IQR) | Frequency (%) |
|---|---|---|
| **Age(years)** | | |
| | 38 (29–49) | |
| **Sex** | | |
| Male | | 189 (94.00) |
| Female | | 12 (6.00) |
| **Educational status** | | |
| Primary school | | 63 (31.30) |
| Secondary school | | 111 (55.20) |
| Tertiary institution | | 27 (13.40) |

IQR: InterQuartile Range; n = sample size

**Table 2. Descriptive summary of exposure to occupational biopsychosocial factors.**

| Variable; n = 201 | Category | Frequency (%) | Mean (SD) | Median (IQR) |
|---|---|---|---|---|
| | | **Summary of exposure to occupational biomechanical factors** | | |
| **Head load carriage history** | | | | |
| Weight of load (kilogram) | | | 39.26 (12.09) | |
| Frequency of load carriage (number of days per week) | | | | 6.00 (5–6) |
| Duration of load carriage (number of hours per day) | | | | 9.00 (8–10) |
| **Neck posture history** | | | | |
| Neck in predominantly forward posture | | 7.00 (3.50) | | |
| Neck in predominantly extended posture | | 4.00 (2.00) | | |
| Neck in predominantly twisted(rotated) posture | | 190.00 (94.50) | | |
| Frequency of neck in predominant posture (number of days per week) | | | | 6.00 (1.00) |
| Duration of neck in predominant posture (number of hours per day) | | | | 9.00 (2.00) |
| | | **Summary of the occupational psychosocial factors** | | |
| Can you usually take breaks in your job in addition to the scheduled breaks? | Yes<br>No | 31 (15.40)<br>170 (84.60) | | |
| Do you often find that you cannot work because of unexpected events, such as machine breakdown or material not delivered? | Yes<br>No | 193 (96.00)<br>8 (4.00) | | |
| Can you usually control the order and the pace of your tasks? | Yes<br>No | 153 (76.10)<br>48 (23.90) | | |
| Is the order and pace of your tasks usually dependent on others (machines, computers, customers)? | Yes<br>No | 148 (73.60)<br>23 (26.40) | | |

(*Continued*)

**Table 2.** (Continued)

| Variable; n = 201 | Category | Frequency (%) | Mean (SD) | Median (IQR) |
|---|---|---|---|---|
| Do you usually work under time pressure and deadlines? | Yes<br>No | 146 (72.60)<br>55 (27.40) | | |

SD: standard deviation; IQR: interquartile range

these variables (frequency of load carriage and neck posture frequency) were removed from the final regression models to eliminate multicollinearity.

In model 1 of Table 6, the demographic variables (age and education) significantly (p≤0.25) associated with neck pain intensity were entered into the first block of a sequential multiple regression to control their effects. This model explained 12% of the variance in neck pain intensity (adjusted $R^2$ = 0.12), and was significant (F (2, 198) = 14.96, p<0.0001). Model 2, in which all the occupational biomechanical factors were added, explained significantly more variance ($R^2$ change = 0.08, F (5, 195) = 10.47, p<0.0001) with the model explaining 19% of the variance in neck pain intensity (adjusted $R^2$ = 0.19). Model 3, in which all the occupational psychosocial factors were added, explained significantly more variance ($R^2$ change = 0.35, F (10, 190) = 24.01, p<0.0001) with the model explaining 53% of the variance in neck pain intensity (adjusted $R^2$ = 0.53). The significant factors associated with neck pain intensity were 'Is the order and pace of your tasks usually dependent on others (machines, computers, customers)?' (β = 0.35; p<0.0001), 'Can you usually take breaks in your job in addition to the scheduled breaks?' (β = 0.25; p<0.0001), 'Do you often find that you cannot work because of unexpected events, such as machine break down or material not delivered?' (β = 0.21; p<0.0001), Can you usually control the order and pace of your tasks? (β = 0.20; p<0.0001), and weight of load (β = 0.17; p<0.0001) (Table 6).

In model 1 of Table 7, the occupational biomechanical factors were entered into the first block of a sequential multiple regression (no significant socio-demographic factor to control for [p>0.25]). This model explained 16% of the variance in neck disability (adjusted $R^2$ = 0.16), and was significant (F (4, 196) = 10.241, p<0.0001). Model 2, in which all the occupational psychosocial factors were added, explained significantly more variance ($R^2$ change = 0.06; F (9, 191) = 6.474, p<0.0001) with the model explaining 20% of the variance in neck disability (adjusted $R^2$ = 0.20). The significant factors associated with neck disability were weight of load (β = 0.30; p<0.0001), duration of load carriage (β = 0.16; p = 0.01), Do you usually work under time pressure and deadlines? (β = 0.16; p = 0.02), (Table 7).

**Table 3. Descriptive summary of the dependent neck pain outcomes–neck pain intensity, neck disability and sick leave.**

| Variable; n = 201 | Categories | Frequency (%) | Median (IQR) |
|---|---|---|---|
| Neck pain intensity | | | 6.00 (5–8) |
| Neck disability | | | 18.00 (14–24) |
| | None | 0 (0) | |
| | Mild | 53 (26.40) | |
| | Moderate | 98 (48.80) | |
| | Severe | 42 (20.90) | |
| | Complete | 8 (4.00) | |
| Sick leave | | | 1.00 (0–3) |

IQR: interquartile range

**Table 4. Bivariate associations between each of the socio-demographic variables, occupational biomechanical factors, occupational psychosocial factors, and each of the dependent variables (neck pain intensity, neck disability and sick leave).**

| Variable; n = 201 | Neck pain intensity | Neck disability | Sick leave |
|---|---|---|---|
| **Socio-demographic characteristics** | | | |
| $r_s$(P-value) | | | |
| Age (years) | 0.15(0.02) | 0.01(0.89) | 0.03(0.64) |
| U (P-value) | | | |
| Sex | 973.50(0.40) | 0717.50(0.30) | 930.00(0.38) |
| H (p-value) | | | |
| Educational status | -0.14(0.04) | 3.52(0.74) | -0.00(0.55) |
| **Occupational biomechanical factors** | | | |
| *Head load carriage history* | | | |
| Weight of Load (kilogram) | 0.29(0.00) | 0.33(0.00) | 0.05(0.41) |
| Frequency of load Carriage (number of days per week) | 0.20(0.00) | 0.15(0.03) | -0.06(0.35) |
| Duration of load carriage (number of hours per day) | 0.02(0.73) | 0.16 (0.03) | 0.06(0.41 |
| *Neck posture history* | | | |
| Frequency of neck posture (number of days per week) | -0.20(0.00) | 0.09(0.17) | -0.11(0.06) |
| Duration of neck posture (number of hours per day) | -0.04(0.57) | 0.06(0.34) | -0.02(0.97) |
| Neck forward posture | -0.17 (0.02) | -0.16 (0.02) | -0.05 (0.46) |
| Neck backward posture | -0.15 (0.04) | -0.14 (0.05) | -0.05 (0.50) |
| Neck twisted (rotated) posture | 0.21(0.00) | 0.20(0.00) | 0.06(0.35) |
| H (p-value) | | | |
| Predominant neck posture | 11.82 (0.00) | 4.82 (0.01) | 0.52 (0.59) |
| **Occupational psychosocial factors** | | | |
| U (p-value) | | | |
| Can you usually take breaks in your job in addition to the scheduled breaks? | 991.00 (0.00) | 2567.50 (0.82) | 2401.500 (0.41) |
| Do you often find that you cannot work because of unexpected events, such as machine break down or material not delivered? | 99.50 (0.00) | 350.50 (0.01) | 615.500 (0.31) |
| Can you usually control the order and the pace of your tasks? | 1831. 50 (0.00) | 2666.50 (0.00) | 3278.00 (0.24) |
| Is the order and pace of your tasks usually dependent on others (machines, computers, customers)? | 1156.50 (0.00) | 2802.00 (0.00) | 3634.500 (0.40) |
| Do you usually work under time pressure and deadlines? | 2348.50 (0.00) | 2534.50 (0.00) | 3571.500 (0.20) |

$r_s$ = Spearman's correlation; U: Mann-Whitney U test; H: Kruskall Wallis test

In model 1 of Table 8, the occupational biomechanical factors were entered into the first block of a sequential multiple regression (no significant socio-demographic factor to control for [p>0.25]). This model explained 1% of the variance in sick leave (adjusted $R^2$ = 0.01), and was not significant (F (4, 196) = 1.522, p = 0.197). Model 2, in which all the occupational psychosocial factors were added, explained less variance which was not significant ($R^2$ change = 0.01; F (9, 191) = 0.909, p = 0.519) with the model explaining -4% of the variance in sick leave (adjusted $R^2$ = -0.04). The significant factor associated with sick leave was duration of load carriage (β = 0.15; p = 0.04) (Table 8).

Tables 9 and 10 show the sequential multiple linear regression analyses involving neck pain intensity with neck disability and sick leave as the criterion variables respectively. In model 1 of Table 9, the occupational biomechanical factors were entered into the first block of a sequential multiple regression (no significant socio-demographic factor [p>0.25] to control

**Table 5. Bivariate correlation matrix.**

| Variables | 2 | 3 | 4 | 5 | 6 | 7 | 8 | 9 | 10 | 11 | 12 | 13 |
|---|---|---|---|---|---|---|---|---|---|---|---|---|
| | | | | | | | | | | | | |
| 1.Weight of load (kilogram) | 0.15 (0.026) | -0.02 (0.75) | - 0.06 (0.37) | - 0.14 (0.04) | 0.13 (0.05) | 0.13 (0.06) | -0.02 (0.76) | 0.55 (0.41) | 0.15 (0.03) | 0.11 (0.10) | 0.11 (0.10) | 0.05 (0.44) |
| 2.Frequency of load carriage (number of days per week) | – | 0.25 (0.00) | -0.06 (0.38) | - 0.04 (0.56) | 0.07 (0.28) | 0.89 (0.00) | 0.23 (0.00) | 0.07 (0.31) | 0.10 (0.15) | 0.14 (0.04) | -0.04 (0.95) | 0.18 (0.00) |
| 3.Duration of load carriage (number of hours per day) | – | – | 0.05 (0.46) | 0.07 (0.32) | - 0.88 (0.24) | 0.17 (0.01) | 0.70 (0.00) | 0.01 (0.87) | 0.14 (0.04) | 0.05 (0.44) | 0.08 (0.23) | 0.06 (0.33) |
| 4.Forward neck posture | – | – | – | - 0.27 (0.70) | -0.78 (0.00) | -0.05 (0.48) | 0.07 (0.34) | -0.19 (0.00) | -0.19 (0.00) | -0.23 (0.00) | -0.08 (0.25) | -0.11 (0.13) |
| 5.Extended neck posture | – | – | – | – | -0.59 (0.00) | -0.03 (0.61) | 0.02 (0.50) | -0.23 (0.00) | -0.16 (0.02) | 0.03 (0.68) | -0.06 (0.40) | -0.08 (0.26) |
| 6.Twisted (rotated) neck posture | – | – | – | – | – | 0.06 (0.37) | -0.09 (0.23) | 0.29 (0.00) | 0.25 (0.00) | 0.18 (0.01) | 0.10 (0.14) | 0.14 (0.06) |
| 7.Neck posture frequency (number of days per week) | – | – | – | – | – | – | 0.28 (0.00) | 0.11 (0.13) | 0.09 (0.19) | 0.14 (0.06) | 0.01 (0.84) | 0.14 (0.05) |
| 8.Neck posture duration (number of hours per day) | – | – | – | – | – | – | – | 0.00 (0.98) | 0.13 (0.07) | 0.03 (0.64) | 0.11 (0.14) | -0.02 (0.80) |
| 9.Do you usually work under time pressure and deadlines? | – | – | – | – | – | – | – | – | 0.26 (0.00) | 0.22 (0.00) | 0.11 (0.13) | 0.21 (0.00) |
| 10.Is the order and pace of your tasks usually dependent on others (machines, computers, customers)? | – | – | – | – | – | – | – | – | – | 0.18 (0.01) | 0.16 (0.02) | 0.20 (0.00) |
| 11.Do you often find that you cannot work because of unexpected events, such as machine break down or material not delivered? | – | – | – | – | – | – | – | – | – | – | 0.09 (0.22) | 0.11 (0.10) |
| 12. Can you usually take breaks in your job in addition to the scheduled breaks? | – | – | – | – | – | – | – | – | – | – | – | 0.02 (0.79) |
| 13.Can you usually control the order and pace of your tasks? | – | – | – | – | – | – | – | – | – | – | – | – |

$r_s$ (p-value)
$r_s$ = Spearman ranking correlation was used for all variables. Categorical (binary) variables were transformed into dummy variables.

Tables 6–8 show the sequential multiple linear regression analyses with the three criterion variables–neck pain intensity, neck disability, and sick leave respectively.

for). This model explained 16% of the variance in neck disability (adjusted $R^2 = 0.16$), and was significant ($F_{(4, 196)} = 10.241$, p<0.0001). Model 2, in which all the occupational psychosocial factors were added, explained significantly more variance ($R^2$ change = 0.06; $F_{(9, 191)} = 6.474$, p<0.0001) with the model explaining 20% of the variance in neck disability (adjusted $R^2 = 0.20$). Model 3, in which pain intensity was added, explained significantly more variance ($R^2$ change = 0.11; $F_{(10, 190)} = 10.017$, p<0.0001) with the model explaining 31% of the variance in neck disability (adjusted $R^2 = 0.31$). The significant factors associated with neck disability after accounting for pain intensity were neck pain intensity ($\beta = 0.52$; p< 0.0001), weight of load ($\beta = 0.22$; p< 0.0001), Can you usually take breaks in your job in addition to the scheduled breaks? ($\beta = -0.22$; p< 0.0001), duration of load carriage ($\beta = 0.20$; p< 0.0001), Is the order and pace of your tasks usually dependent on others (machines, computers, customers)? ($\beta = 0.15$; p = 0.03) (Table 9).

In model 1 of Table 10, the occupational biomechanical factors were entered into the first block of a sequential multiple regression (no significant socio-demographic factor [p≤0.25] to control for). This model explained 1% of the variance in sick leave (adjusted $R^2 = 0.01$), and was not significant ($F_{(4, 196)} = 1.522$, p = 0.197). Model 2, in which all the occupational psychosocial factors were added, explained less variance which was not significant ($R^2$ change = 0.01; $F_{(9, 191)} = 0.909$, p = 0.519) with the model explaining -4% of the variance in

**Table 6. Sequential multiple regression analysis with neck pain intensity as the criterion variable.**

| Variable, n = 201 | Model 1 | | | Model 2 | | | Model 3 | | |
|---|---|---|---|---|---|---|---|---|---|
| | B (95% Cl) | SEB | β (p value) | B (95% Cl) | SEB | β (p value) | B (95% Cl) | SEB | β (p value) |
| Age | 0.01(-0.02–0.05) | 0.02 | 0.07 (0.43) | 0.02(-0.01–0.05) | 0.02 | 0.12(0.13) | 0.01 (-0.02–0.03) | 0.01 | 0.03(0.63) |
| Education (primary school) (Others = reference) | 0.73(-0.12–1.59) | 0.44 | 0.14 (0.09) | 0.40(-0.39–1.19) | 0.40 | 0.08(0.32) | 0.27(-0.34–0.88) | 0.31 | 0.05(0.38) |
| Weight of load (kilogram) | | | | 0.06(0.03–0.09) | 0.02 | 0.28 ((<0.0001) | 0.04(0.02–0.06) | 0.01 | 0.17 ((<0.0001) |
| Duration of load carriage (number of hours per day) | | | | 0.05(-0.14–0.23) | 0.09 | 0.03(0.60) | -0.08(-0.23–0.06) | 0.07 | -0.06(0.25) |
| Twisted neck posture (yes) (No = reference) | | | | 2.95(1.62–4.27) | 0.67 | 0.28 ((<0.0001) | 0.88(-0.20–1.96) | 0.55 | 0.09(0.11) |
| Neck posture duration (number of hours per day) | | | | 0.01(-0.04–0.05) | 0.02 | 0.02(0.75) | 0.01(-0.03–0.05) | 0.02 | 0.03(0.56) |
| Taking additional breaks (No) (Yes = reference) | | | | | | | 1.60(0.96–2.24) | 0.32 | 0.25 (<0.0001) |
| Unexpected events (Yes) (No = reference) | | | | | | | 2.50(1.29–3.70) | 0.61 | 0.21 (<0.0001) |
| Order and pace control (No) (Yes = reference) | | | | | | | 1.12(0.56–1.67) | 0.28 | 0.20 (<0.0001) |
| Task dependency (yes) (No = reference) | | | | | | | 1.88(1.31–2.45) | 0.29 | 0.35 (<0.0001) |
| Time pressure (yes) (No = reference) | | | | | | | 0.52(-0.03–1.08) | 0.28 | 0.10(0.07) |
| $R^2$ | 0.13 | | | 0.21 | | | 0.56 | | |
| $R^2$ change | 0.13 | | | 0.08 | | | 0.35 | | |
| Adjusted $R^2$ | 0.12 | | | 0.19 | | | 0.53 | | |
| F for change in $R^2$ | $F_{(2, 198)}$ = 14.96 (<0.0001) | | | $F_{(5, 195)}$ = 10.47 (<0.0001) | | | $F_{(10, 190)}$ - 24.01 (<0.0001) | | |

B-unstandardised beta, β-standardised beta, SEB-Standard error of beta. B-unstandardised beta, β-standardised beta, SEB-Standard error of beta. Taking additional breaks = Can you usually take breaks in your job in addition to the scheduled breaks? Unexpected events = Do you often find that you cannot work because of unexpected events, such as machine break down or material not delivered? Order and pace control = Can you usually control the order and pace of your tasks? Task dependency = Is the order and pace of your tasks usually dependent on others (machines, computers, customers)? Time pressure = Do you usually work under time pressure and deadlines?

sick leave (adjusted $R^2$ = -0.04). Model 3, in which pain intensity was added, explained less variance ($R^2$ change = 0.004; $F_{(10, 190)}$ = 0.902, p = 0.533) with the model explaining -5% of the variance in sick leave (adjusted $R^2$ = -0.05). The significant factor associated with sick leave after accounting for pain intensity was duration of load carriage (β = 0.16; p = 0.04 (Table 10).

## Discussion

This study investigated the occupational biopsychosocial factors associated with multiple neck pain outcomes (neck pain intensity, functional disability, and sick leave) amongst construction labourers in an urban African population.

The results aligned with the proposed biopsychosocial theoretical model, except for sick leave which had a statistically non-significant regression model. The significant independent factors associated with neck pain intensity were order and pace of tasks usually being dependent on others (e.g., machines, computers, customers), usually unable to take breaks in the job in addition to the scheduled breaks, cannot work because of unexpected events such as

**Table 7. Sequential multiple regression analysis with neck disability as the criterion variable.**

| Variable, n = 201 | Model 1 | | | Model 2 | | |
|---|---|---|---|---|---|---|
| | B (95% Cl) | SEB | β (p value) | B (95% Cl) | SEB | β (p value) |
| Weight of load (kilogram) | 0.20(0.12–0.28) | 0.04 | 0.32((<0.0001) | 0.19(0.12–0.27) | 0.04 | 0.30((<0.0001) |
| Duration of load carriage (number of hours per day) | 0.71(0.18–1.24) | 0.27 | 0.18(0.01) | 0.66(0.13–1.19) | 0.27 | 0.16(0.01) |
| Twisted neck posture (yes) (No = reference) | 5.42(1.60–9.24) | 1.94 | 0.19(0.01) | 3.34(-0.64–7.32) | 2.02 | 0.11(0.10) |
| Neck posture duration (number of hours per day) | -0.07(-0.20–0.07) | 0.07 | -0.07(0.32) | -0.07(-0.20–0.06) | 0.07 | -0.07(0.29) |
| Taking additional breaks (No) (Yes = reference) | | | | -1.60(-3.96–0.76) | 1.20 | -0.09(0.18) |
| Unexpected events (Yes) (No = reference) | | | | 1.29(-3.15–5.73) | 2.25 | 0.04(0.57) |
| Order and pace control (No) (Yes = reference) | | | | 2.03(-0.02–4.07) | 1.04 | 0.13(0.05) |
| Task dependency (yes) (No = reference) | | | | 0.57(-1.50–2.64) | 1.05 | 0.04(0.59) |
| Time pressure (yes) (No = reference) | | | | 2.41(0.36–4.46) | 1.04 | 0.16(0.02) |
| $R^2$ | 0.17 | | | 0.23 | | |
| $R^2$ change | 0.17 | | | 0.06 | | |
| Adjusted $R^2$ | 0.16 | | | 0.20 | | |
| F for change in $R^2$ | F (4, 196) = 10.241 (<0.0001) | | | F (9, 191) = 6.474 (<0.0001) | | |

B-unstandardised beta, β-standardised beta, SEB-Standard error of beta. B-unstandardised beta, β-standardised beta, SEB-Standard error of beta. Taking additional breaks = Can you usually take breaks in your job in addition to the scheduled breaks? Unexpected events = Do you often find that you cannot work because of unexpected events, such as machine break down or material not delivered? Order and pace control = Can you usually control the order and pace of your tasks? Task dependency = Is the order and pace of your tasks usually dependent on others (machines, computers, customers)? Time pressure = Do you usually work under time pressure and deadlines?

machine break down or material not delivered, usually unable to control the order and the pace of tasks, and the weight of load (kilograms). These factors included all the psychosocial factors (except usually working under time pressure and deadlines), and one biomechanical factor, and together explained 54.0% of the variance in neck pain intensity.

Occupational psychosocial factors appeared to be the most important independent factors associated with neck pain intensity with all but one of the questionnaire items (usually working under time pressure and deadlines) significantly associated with neck pain intensity. This concurs with evidence in high income countries [31, 63, 64]. Weight of load was also significantly associated with neck pain intensity but did not appear to be as important as the psychosocial factors. This agrees with evidence from both high and lower income countries [65, 66]. In contradiction, a study in a high income country that also included physiological factors found that negative affectivity, greater neck flexor activity during cranio-cervical flexion, and longer duration of symptoms, but not work-related psychosocial factors were predictors of neck pain [67]. However, that study may have been underpowered to predict the large number of variables included. Moreover, the presence of neck pain which was the construct measured in that study is different from the intensity of neck pain assessed in the present study [67]. The comparatively higher predictive power of the regression model for pain intensity (54%) in this present study implies the inclusion of relatively more relevant factors associated with pain intensity.

The significant independent factors associated with neck disability were weight of load (kilograms), usually working under time pressure and deadlines, and duration of load carriage (number of hours per day). These factors included two biomechanical factors and one psychosocial factor which together explained 20.0% of the variation in neck disability. All except one of the occupational psychosocial factors (usually unable to take breaks in the job in addition to

**Table 8. Sequential multiple regression analysis with sick leave as the criterion variable.**

| Variable, n = 201 | Model 1 | | | Model 2 | | |
|---|---|---|---|---|---|---|
| | B (95% Cl) | SEB | β (p value) | B (95% Cl) | SEB | β (p value) |
| Weight of load (kilogram) | 0.01(-0.03–0.04) | 0.02 | 0.02(0.75) | 0.00(-0.03–0.03) | 0.02 | 0.01(0.87) |
| Duration of load carriage (number of hours per day) | 0.21(0.01–0.40) | 0.10 | 0.15(0.04) | 0.23(0.01–0.41) | 0.10 | 0.15(0.04) |
| Twisted neck posture (yes) (No = reference) | 0.84(-0.56–2.25) | 0.71 | 0.09(0.24) | 0.53(-0.98–2.04) | 0.77 | 0.05(0.49) |
| Neck posture duration (number of hours per day) | -0.03(-0.08–0.02) | 0.03 | -0.08(0.27) | -0.03(-0.08–0.02) | 0.03 | -0.08(0.26) |
| Taking additional breaks (No) (Yes = reference) | | | | 0.31(-0.59–1.20) | 0.45 | 0.05(0.50) |
| Unexpected events (Yes) (No = reference) | | | | 0.11(-1.57–1.80) | 0.85 | 0.01(0.90) |
| Order and pace control (No) (Yes = reference) | | | | 0.01(-0.77–0.79) | 0.39 | 0.00(0.98) |
| Task dependency (yes) (No = reference) | | | | -0.02(-0.81–0.77) | 0.40 | -0.00(0.96) |
| Time pressure (yes) (No = reference) | | | | 0.46(-0.32–1.24) | 0.40 | 0.09(0.24) |
| $R^2$ | 0.03 | | | 0.04 | | |
| $R^2$ change | 0.03 | | | 0.01 | | |
| Adjusted $R^2$ | 0.01 | | | -0.04 | | |
| F for change in $R^2$ | F (4, 196) = 1.522 (0.197) | | | F (9, 191) = 0.909 (0.519) | | |

B-unstandardised beta, β-standardised beta, SEB-Standard error of beta. B-unstandardised beta, β-standardised beta, SEB-Standard error of beta. Taking additional breaks = Can you usually take breaks in your job in addition to the scheduled breaks? Unexpected events = Do you often find that you cannot work because of unexpected events, such as machine break down or material not delivered? Order and pace control = Can you usually control the order and pace of your tasks? Task dependency = Is the order and pace of your tasks usually dependent on others (machines, computers, customers)? Time pressure = Do you usually work under time pressure and deadlines?

the scheduled breaks) had significant bivariate associations with neck disability. However, the only occupational psychosocial factor (usually working under time pressure and deadlines) that retained statistical significance in the neck disability regression model was the only occupational psychosocial factor which was not associated with neck pain intensity in the neck pain intensity regression model. The addition of neck pain intensity as an independent variable to the neck disability regression model explained significantly more variation in neck disability (31.0%), with neck pain intensity having the strongest association with neck disability. Weight of load and the duration of load carriage consistently explained neck disability in the two models (with and without pain intensity as an independent variable). However, the addition of pain intensity as an independent variable to the neck disability regression model changed the occupational psychosocial factors associated with neck disability. 'Usually unable to take breaks in the job in addition to the scheduled breaks', and 'order and pace of tasks usually being dependent on others (e.g., machines, computers, customers)', which were not previously associated with neck disability (in the neck disability regression model) but were the two factors with the strongest associations with neck pain intensity (in the neck pain intensity regression model), became the only occupational psychosocial factors associated with neck disability. These results suggest that these occupational psychosocial factors are strongly associated with neck pain intensity.

Pain intensity was the strongest independent factor associated with neck disability, which agrees with previous finding amongst people with back pain in Nigeria [8]. Without pain intensity as an independent variable in the regression model, weight of load was the strongest independent factor associated with neck disability and was significantly associated with neck pain intensity. This aligns with evidence in other African populations showing that increasing

**Table 9. Sequential multiple regression analysis involving neck pain intensity with neck disability as the criterion variable.**

| Variable, n = 201 | Model 1 | | | Model 2 | | | Model 3 | | |
|---|---|---|---|---|---|---|---|---|---|
| | B (95% Cl) | SEB | β (p value) | B (95% Cl) | SEB | β (p value) | B (95% Cl) | SEB | β (p value) |
| Weight of load (kilogram) | 0.20(0.12–0.28) | 0.04 | 0.32 (<0.0001) | 0.19(0.11–0.27) | 0.04 | 0.30 (<0.0001) | 0.14(0.06–0.21) | 0.04 | 0.22 (<0.0001) |
| Duration of load carriage (number of hours per day) | 0.71(0.18–1.24) | 0.27 | 0.18(0.01) | 0.66(0.13–1.19) | 0.27 | 0.16(0.01) | 0.79(0.30–1.27) | 0.25 | 0.20 (<0.0001) |
| Twisted neck posture (yes) (No = reference) | 5.42(1.60–9.24) | 1.94 | 0.19(0.01) | 3.34(-0.64–7.32) | 2.02 | 0.11(0.10) | 2.07(-1.62–5.76) | 1.87 | 0.07(0.27) |
| Neck posture duration (number of hours per day) | -0.07(-0.20–0.07) | 0.07 | -0.07(0.32) | -0.07(-0.20–0.06) | 0.07 | -0.07(0.29) | -0.09(-0.21–0.03) | 0.06 | -0.09(0.14) |
| Taking additional breaks (No) (Yes = reference) | | | | -1.60(-3.96–0.76) | 1.20 | -0.09(0.18) | -3.96(-6.26- -1.65) | 1.17 | -0.22 (<0.0001) |
| Unexpected events (Yes) (No = reference) | | | | 1.29(-3.15–5.73) | 2.25 | 0.04(0.57) | -2.50(-6.77–1.77) | 2.17 | -0.07(0.25) |
| Order and pace control (No) (Yes = reference) | | | | 2.03(-0.02–4.07) | 1.04 | 0.13(0.05) | 0.42(-1.54–2.37) | 0.99 | 0.03(0.67) |
| Task dependency (yes) (No = reference) | | | | 0.57(-1.50–2.64) | 1.05 | 0.04(0.59) | -2.33(-4.46- -0.19) | 1.08 | 0.15(0.03) |
| Time pressure (yes) (No = reference) | | | | 2.41(0.36–4.46) | 1.04 | 0.16(0.02) | 1.65(-0.26–3.55) | 0.97 | 0.11(0.09) |
| Neck pain intensity | | | | | | | 1.47(0.98–1.96) | 0.25 | 0.52 (<0.0001) |
| $R^2$ | 0.17 | | | 0.23 | | | 0.35 | | |
| $R^2$ change | 0.17 | | | 0.06 | | | 0.11 | | |
| Adjusted $R^2$ | 0.16 | | | 0.20 | | | 0.31 | | |
| F for change in $R^2$ | $F_{(4, 196)}$ = 10.241 (<0.0001) | | | $F_{(9, 191)}$ = 6.474 (<0.0001) | | | $F_{(10, 190)}$ = 10.017 (<0.0001) | | |

B-unstandardised beta, β-standardised beta, SEB-Standard error of beta. Taking additional breaks = Can you usually take breaks in your job in addition to the scheduled breaks? Unexpected events = Do you often find that you cannot work because of unexpected events, such as machine break down or material not delivered? Order and pace control = Can you usually control the order and pace of your tasks? Task dependency = Is the order and pace of your tasks usually dependent on others (machines, computers, customers)? Time pressure = Do you usually work under time pressure and deadlines?

load-carrying is associated with increasing neck disability [65]. Duration of load carriage was another significant occupational biomechanical factor, showing statistically significant associations with neck disability, and was the only factor significantly associated with sick leave, but had no association with neck pain intensity. This also concurs with evidence in low and middle-income countries suggesting that the daily duration of exposure to load-carrying is associated with neck disability [65, 68].

Our results contradict results in a high income country showing no consistent associations between physical and psychological job demands and clinically meaningful improvements in neck disability [69]. The contradicting results could be due to different population characteristics and study designs. For instance, the previous study [69] was conducted amongst primary care patients in the USA with potentially lower exposure levels to physical and psychological risk factors in the workplace. This is possibly due to strict occupational health and safety regulations in high income countries which is nearly non-existent in Nigeria. Another reason for the different results could be the potentially less severe implication of job loss to the patients in the USA who may have access to a social welfare system. In contrast, participants in this current study have no similar access to an alternative means of livelihood and survival through a social welfare system. Finally, in contrast to this present observational study that investigated the independent factors associated with neck disability, the USA study [69] was an

**Table 10. Sequential multiple regression analysis involving neck pain intensity with sick leave as the criterion variable.**

| Variable, n = 201 | Model 1 | | | Model 2 | | | Model 3 | | | | |
|---|---|---|---|---|---|---|---|---|---|---|---|
| | B (95% Cl) | SEB | β (p value) | B (95% Cl) | SEB | β (p value) | B (95% Cl) | SEB | β (p value) | | |
| Weight of load (kilogram) | 0.01(-0.03–0.04) | 0.02 | 0.02 (0.75) | 0.00(-0.03–0.03) | 0.02 | 0.01 (0.87) | 0.00(-0.03–0.03) | 0.02 | 0.01 (0.92) | | |
| Duration of load carriage (number of hours per day) | 0.21(0.01–0.40) | 0.10 | 0.15 (0.04) | 0.21(0.01–0.41) | 0.10 | 0.15 (0.04) | 0.21(0.01–0.41) | 0.10 | 0.16 (0.04) | | |
| Twisted neck posture (yes) (No = reference) | 0.84(-0.56–2.25) | 0.71 | 0.09 (0.24) | 0.53(-0.98–2.04) | 0.77 | 0.05 (0.49) | 0.51(-1.01–2.04) | 0.77 | 0.05 (0.51) | | |
| Neck posture duration (number of hours per day) | -0.03(-0.08–0.02) | 0.03 | -0.08 (0.27) | -0.03(-0.08–0.02) | 0.03 | -0.08 (0.26) | -0.03(-0.08–0.02) | 0.03 | -0.08 (0.26) | | |
| Taking additional breaks (No) (Yes = reference) | | | | 0.31(-0.59–1.20) | 0.45 | 0.05 (0.50) | 0.27(-0.69–1.22) | 0.48 | 0.04 (0.58) | | |
| Unexpected events (Yes) (No = reference) | | | | 0.11(-1.57–1.80) | 0.85 | 0.01 (0.90) | 0.05(-1.72–1.82) | 0.90 | 0.00 (0.96) | | |
| Order and pace control (No) (Yes = reference) | | | | 0.01(-0.77–0.79) | 0.39 | 0.00 (0.98) | -0.02(-0.83–0.79) | 0.41 | -0.00 (0.97) | | |
| Task dependency (yes) (No = reference) | | | | -0.02(-0.81–0.77) | 0.40 | -0.00 (0.96) | -0.07(0.95–0.81) | 0.45 | -0.01 (0.88) | | |
| Time pressure (yes) (No = reference) | | | | 0.46(-0.32–1.24) | 0.40 | 0.09 (0.24) | 0.45(-0.34–1.24) | 0.40 | 0.09 (0.26) | | |
| Neck pain intensity | | | | | | | 0.03(-0.18–0.23) | 0.10 | 0.03 (0.81) | | |
| $R^2$ | 0.03 | | | 0.04 | | | 0.05 | | | | |
| $R^2$ change | 0.03 | | | 0.01 | | | 0.004 | | | | |
| Adjusted $R^2$ | 0.01 | | | -0.04 | | | -0.05 | | | | |
| | $F_{(4, 196)} = 1.522$ (0.197) | | | $F_{(9, 191)} = 0.909$ (0.519) | | | $F_{(10, 190)} = 0.902$ (0.533) | | | | |

B-unstandardised beta, β-standardised beta, SEB-Standard error of beta. B-unstandardised beta, β-standardised beta, SEB-Standard error of beta. Taking additional breaks = Can you usually take breaks in your job in addition to the scheduled breaks? Unexpected events = Do you often find that you cannot work because of unexpected events, such as machine break down or material not delivered? Order and pace control = Can you usually control the order and pace of your tasks? Task dependency = Is the order and pace of your tasks usually dependent on others (machines, computers, customers)? Time pressure = Do you usually work under time pressure and deadlines?

intervention study which aimed at determining whether job demands influenced the clinical outcomes of treatment. Moreover, a different outcome tool and scoring criteria were utilized in that study to measure job demands [69]. An observational study in another high income country showed associations between other occupational psychosocial factors including lower workplace social support and job satisfaction, and severity of neck disability [70].

The lower predictive power of the model for neck disability could be due to the non-inclusion of other potentially important independent factors associated with neck disability into the regression model. Non-occupational factors may have been important in explaining neck disability in this population. For instance, illness perceptions, catastrophizing, fear avoidance beliefs, and anxiety which predicted self-reported back pain disability in this population [8]; and self-efficacy [71], duration of pain [67, 72], size of painful areas [73], quality of life and sleep [74], concurrent back pain or shoulder pain [75], emotional distress [76], stress [68, 75], vigorous leisure-time physical activity [70], which were associated with neck disability in other populations; may have been relevant in this population.

Unexpectedly, there was nearly no reported sick leave, and this could explain the surprising finding that none of the two regression models with sick leave as the criterion variable (with or

without pain intensity as an independent variable) was statistically significant. Surprisingly, the regression models with sick leave as the criterion variable reduced from explaining 1% of the variation in sick leave with the occupational biomechanical factors as independent variables, to explaining -4% of the variation in sick leave with the addition of occupational psychosocial factors to explaining -5% of the variation in sick leave with the addition of pain intensity, suggesting that occupational biomechanical factors were the most important factors in explaining sick leave. The duration of load carriage remained the only factor that reached statistical significance (p = 0.04) within the two non-significant regression models predicting sick leave (Tables 8 and 10). These results suggest that despite the very low levels of sick leave reported, occupational biomechanical factors may be more important than occupational psychosocial factors and neck pain intensity in explaining sick leave.

The very low level of sick leave found in this study contradicts the evidence in high-income countries [77–80]. This could be due to participants' low paying informal self-employment meaning that they had to work every day irrespective of neck pain to earn a living in a country with no social benefit system. This partly supports findings that self-employment in high income countries was associated with a lower risk of sick leave [81]. Another reason for the very limited relevance of sick leave in this study could be due to the possible presence of the healthy worker effect. As data collection happened at the construction sites rather than at participants' homes, people who were on sick leave during data collection would not have been captured. However, sick leave was measured by asking participants how many days in the past four weeks they had stayed off from work due to neck pain. This was expected to have counteracted the healthy worker effect as the period captured the past rather than the present. Despite this, it is possible that recall bias [23, 82, 83] was present with the participants potentially answering the question in line with their present work status on the day that data were collected. Therefore, the best ways of measuring the construct of sick leave and the broader construct of work-related disability to minimize bias need further exploration in this population. Similar to the previous population-based cross-sectional study in rural Nigeria involving mostly farmers [8], the participants in this urban-based study in Nigeria likely had similar levels of exposure to occupational biomechanical factors from construction work. A cross-sectional study amongst adolescents in Brazil found that not being in work was a protective factor associated with acute LBP but not chronic LBP [84].

## Strengths and limitations

The strength of this study includes its novelty involving a rarely studied population in a lower middle-income country setting, and the confidence in the estimates due to the robust sample size. However, this study has several limitations. The cross-sectional design of this study constrains the establishment of causality. The involvement of mostly male participants due to the nature of the occupation where very few females are employed in this occupation in Nigeria limits the generalizability of findings. This sample may not be representative of the entire Nigerian population which is multi-state and multi-cultural. The use of self-reported measures increased the risk of recall bias [23, 82, 83]. The use of a work site-based study design increased the risk of the heathy worker effect that probably obfuscated the impact of sick leave in this population. The modest prediction accuracy of the regression models with neck disability and sick leave as the criterion variables suggest the existence of other more important factors.

## Implications for practice and policy

Occupational health regulators need to address the occupational psychosocial and occupational biomechanical factors associated with adverse neck pain outcomes in Nigeria. The

occupational psychosocial factors increase work pressure and reduce the ability of the construction labourers to control aspects of their work. These include usually working under time pressure and deadlines, the order and pace of tasks usually being dependent on external factors such as machines, computers, customers etc, often being unable to work because of unexpected events, such as machine break down or material not delivered, usually unable to take breaks in addition to the scheduled breaks at work, and usually unable to control the order and pace of tasks at work. The occupational biomechanical factors associated with adverse neck pain outcomes were weight of load and duration of load carriage. Occupational health regulators in Nigeria may need to enforce employers' compliance with international labour standards on occupational safety and health including the need for workers to have some autonomy in the workplace, implementation of economic compensation for sick and injured workers in line with the international labour organization's guidelines, and implementation of a maximum weight to be lifted at a time and a maximum number of hours that the weight can be lifted per day, to align with evidence-based occupational health and safety regulations. Finally, the findings from this study may be a call for occupational health regulators to facilitate the mechanization of the construction industry in Nigeria, which has been the case in high income countries.

## Implications for future research

Future research may need to utilize longitudinal study designs with much larger sample sizes to test a greater number of relevant biopsychosocial factors. These studies may need to utilize more sophisticated statistical analyses such as structural equation modelling to establish temporal or causal relationships between the occupational biomechanical and psychosocial factors, and the neck pain outcomes. The use of population-based (as opposed to work site-based) study designs can help to reduce the heathy worker effect and recall bias, thereby clarifying the role of sick leave and the factors associated with it in this population. The broader construct of work-related disability may need to be investigated in these studies. Future studies may need to investigate women in Nigeria and other African countries who routinely carry heavy loads on their heads to and from markets and farms, and for daily tasks including carrying water from streams for domestic use such as cooking and laundry.

## Conclusions

Occupational psychosocial factors were associated with neck pain intensity and neck disability, but no occupational psychosocial factor was associated with sick leave. In contrast, occupational biomechanical factors were associated with all the neck pain outcomes including neck pain intensity, neck disability and sick leave. Occupational psychosocial factors appeared to be more important than occupational biomechanical factors in explaining neck pain intensity. Occupational biomechanical factors appeared to be more important than occupational psychosocial factors in explaining neck disability and sick leave. Weight of load was associated with neck pain intensity and neck disability, whereas duration of load carriage was associated with neck disability and sick leave. 39 kilograms was the average weight lifted by the workers at each point in time, for an average of 9 hours per day.

## Supporting information

**S1 Checklist. STROBE statement—checklist of items that should be included in reports of observational studies.**
(DOC)

**S1 Table. Collinearity diagnostics for the regression models.**
(DOCX)

# Acknowledgments

The authors appreciate the participants for freely giving their time, and for the lead labourers who supported study recruitment.

# Author Contributions

**Conceptualization:** Chinonso N. Igwesi-Chidobe.

**Data curation:** Chinonso N. Igwesi-Chidobe, Excellence Effiong, Joseph O. Umunnah, Benjamin C. Ozumba.

**Formal analysis:** Chinonso N. Igwesi-Chidobe, Excellence Effiong, Benjamin C. Ozumba.

**Funding acquisition:** Chinonso N. Igwesi-Chidobe.

**Investigation:** Chinonso N. Igwesi-Chidobe, Joseph O. Umunnah.

**Methodology:** Chinonso N. Igwesi-Chidobe, Excellence Effiong, Benjamin C. Ozumba.

**Project administration:** Chinonso N. Igwesi-Chidobe, Joseph O. Umunnah, Benjamin C. Ozumba.

**Resources:** Chinonso N. Igwesi-Chidobe.

**Software:** Chinonso N. Igwesi-Chidobe.

**Supervision:** Chinonso N. Igwesi-Chidobe, Joseph O. Umunnah, Benjamin C. Ozumba.

**Validation:** Chinonso N. Igwesi-Chidobe, Excellence Effiong, Joseph O. Umunnah.

**Visualization:** Chinonso N. Igwesi-Chidobe.

**Writing – original draft:** Chinonso N. Igwesi-Chidobe, Excellence Effiong, Benjamin C. Ozumba.

**Writing – review & editing:** Chinonso N. Igwesi-Chidobe.

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
