## [Decision Letter · Decision Letter 0]

28 Dec 2023

PONE-D-23-38173Occupational biopsychosocial factors associated with neck pain intensity, neck-disability, and sick leave: a cross-sectional study of construction labourers in an African populationPLOS ONE

Dear Dr. Igwesi-Chidobe,

Thank you for submitting your manuscript to PLOS ONE. After careful consideration, we feel that it has merit but does not fully meet PLOS ONE’s publication criteria as it currently stands. Therefore, we invite you to submit a revised version of the manuscript that addresses the points raised during the review process.

We look forward to receiving your revised manuscript.

Kind regards,

Mohammad Mofatteh, PhD, MPH, MSc, PGCert TLHE, BSc (Hons)

Academic Editor

PLOS ONE

Journal Requirements:

"The lead author was partly supported with personal grant from the 2020 Nature Research Award for Driving Global Impact."

Reviewers' comments:

Reviewer's Responses to Questions

**Comments to the Author**

1. Is the manuscript technically sound, and do the data support the conclusions?

Reviewer #1: Yes

Reviewer #2: Yes

Reviewer #3: Yes

2. Has the statistical analysis been performed appropriately and rigorously? 

Reviewer #1: Yes

Reviewer #2: Yes

Reviewer #3: Yes

3. Have the authors made all data underlying the findings in their manuscript fully available?

Reviewer #1: Yes

Reviewer #2: Yes

Reviewer #3: Yes

4. Is the manuscript presented in an intelligible fashion and written in standard English?

Reviewer #1: Yes

Reviewer #2: Yes

Reviewer #3: Yes

5. Review Comments to the Author

Reviewer #1: The article is interesting and relevant. The authors took great pains to include people who often do not participate easily in scientific research. As a result, interesting insights have emerged. Further, as far as I can judge, the statistics are adequate and the resulting insights are discussed clearly and in detail.

Reviewer #2: Thank you for giving me the opportunity to review this manuscript. The authors investigated the occupational biomechanical and occupational psychosocial factors associated with neck pain intensity, neck disability and sick leave amongst construction labourers in an urban Nigerian population. This study was well designed and the paper was written well.

Reviewer #3: This is an interesting and novel study that is relevant to readers in several fields including occupational health, pain management, musculoskeletal medicine, etc. The writing style is generally clear, and the methodology seems sound.

The authors do acknowledge some limitations. They studied men in the construction industry. In a future study, it would be very important to study women in Nigeria and other African countries who routinely carry heavy loads on their heads to and from markets and for daily tasks such as carrying water or laundry.

The comparisons to high income countries and countries with more expansive health and disability systems are noteworthy.

The article is pretty dense with detail and has a large number of tables. While the analyses in the tables are relevant and likely of interest to readers in their specific subfield, the authors might consider replacing one or two tables with figures that would help engage the more general readers.

6. PLOS authors have the option to publish the peer review history of their article (what does this mean?). If published, this will include your full peer review and any attached files.

Reviewer #1: No

Reviewer #2: **Yes: **Young-Chang Arai

Reviewer #3: No

---

## [Author Response · Author response to Decision Letter 0]

3 Jan 2024

Response to Editor’s comments

1. I have now reformatted the manuscript to meet PLOS ONE's style requirements. Changes are in red text within the manuscript.

2. I have now amended the role of funder statement and included this in the cover letter as well as in the manuscript in line with your suggestions. Changes are in red text. 

3. a. I have now clarified that due to ethical reasons, data sets can be made available upon request to the Health Research Ethics Committee of the University of Nigeria Teaching Hospital (unth.edu.ng) or to the corresponding author. Changes are in red text. 3b. does not apply. I have now updated my Data Availability statement in the submission form accordingly.

4. I have now included captions for my supporting information files at the end of the manuscript and have now updated the in-text citations to match accordingly following the guidelines you provided. 

5. I have now reviewed my reference list and I can confirm that it is now complete. I can confirm that I did not cite papers that have been retracted rather I have cited our recently completed study which is at the final stages of peer review in The Journal of Pain. I have now updated the reference list to reflect this. 

Response to Reviewers’ comments

Thank you to Reviewer 1, Reviewer 2, and Reviewer 3 for the positive comments.

In response to reviewer 3’s further comments,

I have now included that ‘future studies need to investigate women in Nigeria and other African countries who routinely carry heavy loads on their heads to and from markets and farms, and for daily tasks including carrying water from streams for domestic use such as cooking and laundry’ in the implications for future research subsection of the discussion section. 

I have now moved the previous Table 6 to become a supplementary Table renamed S1 Table. The manuscript currently has 2 figures. The other tables are best illustrated in Tabular form to be more informative for future primary and secondary research. All changes are in red text within the manuscript. 

Response to other comments

6. I am happy for the peer review history of this manuscript with PLOS ONE to be published. 

others: I have now aligned the Figures to the journal specifications and have used the Preflight Analysis and Conversion Engine (PACE) digital diagnostic tool, https://pacev2.apexcovantage.com/ following the guidelines that you provided to ensure that this is the case.

---

## [Decision Letter · Decision Letter 1]

20 Mar 2024

Occupational biopsychosocial factors associated with neck pain intensity, neck-disability, and sick leave: a cross-sectional study of construction labourers in an African population

PONE-D-23-38173R1

Dear Dr. Igwesi-Chidobe,

We’re pleased to inform you that your manuscript has been judged scientifically suitable for publication and will be formally accepted for publication once it meets all outstanding technical requirements.

An invoice for payment will follow shortly after the formal acceptance. To ensure an efficient process, please log into Editorial Manager at Editorial Manager® , click the 'Update My Information' link at the top of the page, and double check that your user information is up-to-date. If you have any billing related questions, please contact our Author Billing department directly at authorbilling@plos.org.

Kind regards,

Shahnawaz Anwer, PhD

Academic Editor

PLOS ONE

Additional Editor Comments (optional):

Reviewers' comments:

Reviewer's Responses to Questions

**Comments to the Author**

1. If the authors have adequately addressed your comments raised in a previous round of review and you feel that this manuscript is now acceptable for publication, you may indicate that here to bypass the “Comments to the Author” section, enter your conflict of interest statement in the “Confidential to Editor” section, and submit your "Accept" recommendation.

Reviewer #2: All comments have been addressed

Reviewer #3: All comments have been addressed

2. Is the manuscript technically sound, and do the data support the conclusions?

Reviewer #2: Yes

Reviewer #3: Yes

3. Has the statistical analysis been performed appropriately and rigorously? 

Reviewer #2: Yes

Reviewer #3: Yes

4. Have the authors made all data underlying the findings in their manuscript fully available?

Reviewer #2: Yes

Reviewer #3: Yes

5. Is the manuscript presented in an intelligible fashion and written in standard English?

Reviewer #2: Yes

Reviewer #3: Yes

6. Review Comments to the Author

Reviewer #2: This referee was satisfied with the previous version, so this referee believes that the new version deserves "Accept".

Reviewer #3: The authors have made satisfactory answers to this reviewer's comments and questions from previous review.

No additional questions or concerns.

7. PLOS authors have the option to publish the peer review history of their article (what does this mean?). If published, this will include your full peer review and any attached files.

Reviewer #2: No

Reviewer #3: No

---

## [Editor Report · Acceptance letter]

25 Mar 2024

PONE-D-23-38173R1 

PLOS ONE

Dear Dr. Igwesi-Chidobe, 

I'm pleased to inform you that your manuscript has been deemed suitable for publication in PLOS ONE. Congratulations! Your manuscript is now being handed over to our production team.

Kind regards, 

on behalf of

Dr. Shahnawaz Anwer 

Academic Editor

PLOS ONE